# Ultrasound-Guided Deep Serratus Plane Block in Cat Cadavers (Felis catus): A Description of Dye and Contrast Media Distribution

**DOI:** 10.3390/ani14202978

**Published:** 2024-10-16

**Authors:** Gonzalo Polo-Paredes, Marta Soler, Francisco Gil, Francisco G. Laredo, Amalia Agut, Sara Carrillo-Flores, Eliseo Belda

**Affiliations:** 1Departamento de Medicina y Cirugía Animal, Facultad de Veterinaria, Universidad de Murcia, 30100 Murcia, Spain; gpolo@um.es (G.P.-P.); mtasoler@um.es (M.S.); laredo@um.es (F.G.L.); amalia@um.es (A.A.); 2Hospital Veterinario Universidad de Murcia, 30100 Murcia, Spain; sacarriflores@gmail.com; 3Escuela Internacional de Doctorado de la Universidad de Murcia, Programa en Ciencias Veterinarias, Universidad de Murcia, 30100 Murcia, Spain; 4Departamento de Anatomía y Anatomía Patológica Comparada, Facultad de Veterinaria, Universidad de Murcia, 30100 Murcia, Spain; cano@um.es

**Keywords:** thoracic analgesia, fascial block, feline, locoregional anaesthesia, serratus plane block, ultrasound-guided

## Abstract

The lateral thoracic wall of cats is innervated by the rami cutanei laterales (RCL) emerging from the intercostales nerves from the second to the twelfth thoracic spinal nerve (T). The ultrasound-guided deep serratus plane (DSP) block is a technique that aims to desensitise these RCL to provide analgesia of the lateral wall of the thorax cranial to the 8th rib. This study aimed to assess the distribution of a mixture of dye and contrast media injected in the fascial plane constituted by the lateral aspect of the intercostales externi muscles and the ribs and the medial aspect of the serratus ventralis thoracis muscle. To fulfil this objective, 14 cat cadavers were employed, and a mixture of methylene blue and iopromide 50:50 was injected in the target fascial plane at the laterocaudal aspect of the 5th rib. Cadavers were then subjected to a computed tomographic exam and anatomical dissection. Our results showed staining of the RCL of the intercostales nerves from T2 to T7. These results show that the DSP block could provide analgesia to the lateral aspect of the cranial wall in cats.

## 1. Introduction

The thoracic wall is a musculoskeletal structure that protects vital organs, gives support to the forelimbs, and provides muscular support for ventilation [1]. In cats, it is innervated by eleven pairs of thoracic spinal nerves (T), from the second (T2) to the twelfth (T12) [2]. Once they emerge from the intervertebral space, these nerves promptly divide into the rami dorsales, rami ventrales, and rami communicantes [2]. The rami dorsales are then divided into the rami mediales and rami laterales (RDL), providing innervation to the epaxial region [3]. The rami ventrales continue ventrally as the intercostales nerves, which run between the intercostales interni muscles and the endothoracic fascia. As they run ventrally, they extend a branch to form the rami cutanei laterales (RCL), which provide sensitive innervation to the lateral thoracic wall [4]. Finally, at their ventral end, the intercostales nerves emerge as rami cutanei ventrales to innervate the ventral midline [5].

The serratus plane block is an ultrasound-guided anaesthesia technique that aims to desensitise the RCL, providing analgesia to the cranio-lateral thoracic wall [6]. It was first described in human medicine by Blanco et al. (2013) [7] to provide analgesia for mastectomies. Since then, it has been used to relieve pain in several clinical situations [8], including rib fractures [9,10], thoracoscopic surgeries [11], and thoracotomies [12]. In veterinary medicine, two approaches have been described in dogs: the superficial and the deep serratus plane (DSP) blocks. In the former, the local anaesthetic is administered between the latissimus dorsi (LD) and the serratus ventralis thoracis (SVT) muscles [13]. Meanwhile, in the DSP block [14,15], the injection is performed in the interfascial plane between the SVT and the intercostales externi (IE) muscles/lateral aspect of the ribs. Several applications of this anaesthetic technique, including mastectomies [15], thoracoscopies [16], and persistent ductus arteriosus surgical correction [17,18], have been suggested as part of a multimodal analgesic protocol in dogs. To the authors’ knowledge, only a case report for cranial lobectomy and pericardiectomy using the serratus plane block as multimodal analgesia has been published in cats [19].

This study aims to assess the distribution of a mixture of methylene blue and iopromide injected by the DSP block technique at the 5th rib. We hypothesise that this technique would be feasible in cats and would stain several RCL cranial to the 8th rib.

## 2. Materials and Methods

The present study was approved by the Biosafety Committee in Experimentation (CBE 556/2023) of the University of Murcia (UMU). The cadavers were donated to the UMU by the cadaver donation program of the Faculty of Veterinary Medicine by their owners once the cats died for reasons unrelated to the study. Cadavers were frozen promptly after dead and thawed at room temperature (20–22 °C) 48 h before use.

Cadavers were included in the study if their body condition score (BCS) ranged between 3/9 and 7/9, based on visual and palpable physical signs after thawing, according to the WSAVA Classification [20]. If evidence of trauma/anatomical alteration in the cervical or thoracic area was observed (physical exam and radiography), the cadaver was excluded from the study. A total of 18 cat cadavers were assessed for eligibility.

This study was then divided into two phases:

### 2.1. Phase 1: Anatomical Study

Two cat cadavers were employed to assess the lateral thoracic wall from the first to the thirteenth rib. The lateral and ventral aspects of the neck and thorax were clipped. Then, cadavers were positioned in a supine position. A midline incision was made, and the skin was dissected. The pectoralis superficialis and the pectoralis profundus muscles were detached from their origin at the sternum and lateralised. The RCL of the second to the tenth intercostales nerves were identified between the SVT and the LD. The thoracicus longus nerve was also identified in the lateral aspect of the SVT. The correlation of these nerves with the anatomic structures nearby was assessed. The RCL were then detached from their integration into the LD, and the SVT was dissected from its insertion in the fascia serrata of the scapula. Then, the RCL lying between the IE and the serrations of the SVT were identified, and their relationship with the adjacent anatomical structures was evaluated. All the dissections were made by the same researcher (F.G.).

### 2.2. Phase 2

#### 2.2.1. Ultrasound-Guided Technique

In this phase, 16 feline cadavers were assessed. A total volume of 0.8 mL kg^−1^ of a 50:50 mixture of methylene blue (10 mg mL^−1^, Pancrear Quimica, AppliChem, Castellar del Vallès, Spain) and iopromide (300 mg mL^−1^, UltraVist300, Bayer, Berlin, Germany) was prepared in two syringes (0.4 mL kg^−1^ each) [21,22].

Once clipped as previously described, cadavers were placed in lateral recumbency, and the fifth rib was identified by ultrasound, counting caudocranially from the thirteenth to the fifth rib. A linear ultrasound probe of 3–13 Hz (SL1543, MyLab Gamma, Esaote, Florence, Italy) was positioned caudal to the acromion and transversal to the ribs (Figure 1).

The fourth, fifth, and sixth ribs were identified as echogenic structures producing acoustic shadowing and were used as sonographic landmarks. Then, the LD, SVT, and IE muscles were identified (Figure 2).

A sonovisible needle (Ultraplex 20 G, 100 mm, 30°, BBraun, Melsungen, Germany) was advanced “in plane” in a caudo-cranial direction until the tip of the needle contacted the caudo-lateral aspect of the fifth rib. A volume test of the prepared solution (approximately 0.1 mL) was first injected. When the ultrasound image was compatible with the expected distribution (anechoic pocket between the SVT and the IE/rib), the remaining volume (to reach 0.4 mL kg^−1^) was administered (Figure 3). If the anechoic pocket could not be observed, the injectate was administered based on the ultrasonographic visualisation of the needle tip in contact with the caudo-lateral aspect of the fifth rib. The same procedure was reproduced in the contralateral hemithorax. All the ultrasound procedures were performed by the same researcher (G.P.-P.). Images and videos from the US technique were stored and re-evaluated after the overall procedure was performed.

#### 2.2.2. Computed Tomography (CT) Study

After the administration of the injectate (15–20 min), cadavers were subjected to CT scans (dual-slice CT scanner, General Electric HiSpeed, General Electric Healthcare, Madrid, Spain) of the region between the sixth cervical vertebra to the third lumbar vertebra. Images were obtained with the cadavers positioned in dorsal recumbency with their thoracic limbs extended. The collimator pitch was set to 1, the slice thickness to 3 mm, and the reconstruction intervals with 50% overlap were 100 mA and kVp 120. Then, standard bone and soft tissue reconstruction algorithms were employed. Two certified radiologists (M.S. and A.A.) evaluated the location and distribution of the contrast medium in the reformatted images.

#### 2.2.3. Spread Study

Promptly after the CT scan, cadavers were dissected following the procedure described above *(2.1 Phase 1: Anatomical Study)*. Nerves were considered positively dyed when at least 1 cm in length and all their circumference was stained [23]. The number and identity of the RCL stained in both the superficial and the DSP were assessed, as well as the RDL and the thoracicus longus nerve. Finally, a radical unilateral costotomy was performed, and thoracic viscera were evaluated to assess the presence of dyeing. All the dissections were performed by the same researchers (F.G. and G.P.-P.).

### 2.3. Statistical Analysis

A descriptive statistical analysis was performed using Microsoft Excel 365 (Microsoft Corporation, Redmond, WA, USA) and the plug-in software Real Statistics Resource Pack (release 7.6, Copyright 2013–2021, Charles Zaiontz, www.real-statistics.com accessed on 15 January 2024). Normality distribution was assessed by a Shapiro-Wilk test. The results are expressed as mean ± standard deviation or median (range) whether the distribution was normal or non-normal, respectively.

## 3. Results

### 3.1. Phase 1: Anatomical Study

Two European shorthair female cat cadavers weighing 3.75 and 1.75 kg and having a BCS of 4/9 were included in this phase.

The SVT was observed lying medial to the LD, pectoralis profundus, and pectoralis superficialis muscles, and lateral to the IE muscles, from the first to the ninth ribs, and the origins of the obliquus externus abdominis muscle. Motor innervation of the SVT is provided by the long thoracic nerve, which was observed along its lengthy path on the lateral aspect of the muscle. The origins of the SVT were noted as “serrations” emerging from the first to the ninth rib, converging dorsally to insert on the medial surface of the scapula (fascia serrata). The RCL emerged from the caudal aspect of the second to the eighth rib, at the same level as the acromion, traversing a narrow path between the IE and SVT muscles and emerging between the “serrations” of the SVT to integrate into the LD muscle (Figure 4 and Figure 5).

### 3.2. Phase 2

#### 3.2.1. Demographic Distribution

A total of 16 cats were initially included in Phase 2. Nevertheless, four of them were excluded after thawing due to a massive mammary mass in the thoracic region (1/4) and cachexia (3/4). Finally, 12 animals were included in Phase 2: six males and six females. They weighed 3.33 ± 1.30 kg and had a BCS of 4 (3–7) out of 9.

#### 3.2.2. Ultrasound-Guided Technique

The needle path was visualised in all the injections (24/24). In one hemithorax, the injectate was administered in the erector spinae region and was excluded from the study. Therefore, 23 hemithoraces were finally included.

An anechoic pocket was observed following the administration of the injectate in 73.91% (17/23) of the injections. This pocket initially separated the SVT from the IE muscles and quickly disappeared. A certain amount of injectate refluxed through the needle path in 26.09% (6/23) of the injections. Additionally, some degree of intramuscular injection into the SVT was observed in 78.26% (18/23) of the hemithoraces.

#### 3.2.3. Computed Tomography (CT) Study

The contrast media was located in the target plane in 95.65% (22/23) of the hemithoraces. On the one left, it was located in the subcutaneous space. The contrast media spread a median of 5.5 (2.5–7.5) intercostal spaces. Two distinct patterns were observed: in 43.47% (10/23) of the injections, the contrast media pooled in the injection area, while in 56.53% (12/23) of the hemithoraces, it extended dorsally up to the level of the epaxial muscles (Figure 6 and Figure 7).

#### 3.2.4. Spread Study

Dye was found in the target plane in 22/23 hemithoraces and in the subcutaneous space in 1/23. The dye solution was also found in the superficial SVT plane in 18/23 hemithoraces. A median of 3 (0–5) nerves, from RCL2 to RCL7, were stained in the target DSP, whereas a median of 1 (0–5) RCL were found dyed in the superficial plane (Table 1 and Figure 8). In one cadaver, the dye was found in the subcutaneous space in one hemithorax. In the same cadaver, no RCL were found stained in the contralateral hemithorax. Additionally, the thoracicus longus nerve was found stained in 17.39% (4/23) of the hemithoraces. No intrathoracic dye was found.

The methylene blue was also observed dorsally between serratus dorsalis thoracis and the LD muscles, staining the RDL of the rami dorsales of the spinal nerves between T2–T7 (Figure 7). A median of 2 (0–6) RDL were dyed, specifically RDL2 (17.39%), RDL3 (39.13%), RDL4 (43.48%), RDL5 (39.13%), RDL6 (43.48%), and RDL7 (30.43%) (Figure 9).

## 4. Discussion

This study shows that the DSP block performed at the 5th rib is a feasible technique in cats and that 0.4 mL kg^−1^ of a mixture of methylene blue and iopromide regularly stains the RCL from RCL4 to RCL6. These findings are compatible with the supply of analgesia to the cranio-lateral thoracic area innerved by these nerves.

Two approaches, superficial [13] and deep [14,15], have been described to develop the ultrasound-guided serratus plane block in dogs. Despite comparative studies between these two approaches being lacking, according to our observations in cat cadavers, we hypothesise that the DSP block would stain more efficiently the target RCL for two reasons. First, the interfascial plane between IE and SVT (target of the deep approach) seemed smaller than the one between SVT and LD (target of the superficial approach). This fact concentrates on the distribution of the injectate to a smaller area. Second, the RCL were greatly covered by fat in the superficial interfascial plane (SVT-LD), which could decrease the contact of the local anaesthetics with the target nerves.

We decided to use the R5 as the anatomical landmark to administer the injectate because it is in the middle length of the SVT. Additionally, the results could easily be compared with previous studies performed in the superficial and DSP block approaches in dog cadavers [13,14], a species anatomically close to cats. However, we have observed certain anatomical differences in this area between both species. In cat cadavers, the RCL do not pierce the SVT [3] but slide between the serrations of this muscle and continue laterally to reach the LD. Moreover, the SVT in cats extends more caudally (8th to 9th rib) [2,24] than in dogs (7th to 8th rib) [13,14]. Therefore, the DSP block could potentially cover more RCL in cats than in dogs. However, this hypothesis was not supported by our results, as no RCL beyond RCL7 was stained in any hemithorax. The volume of 0.4 mL kg^−1^ per hemithorax was chosen as it is equivalent to a dose of 2 mg kg^−1^ of bupivacaine or ropivacaine 0.5%. Further volume could exceed the recommended doses of these two commonly used local anaesthetics in cats [25,26]. Dilution of a local anaesthetic is a very common practice when a wider area needs to be covered. However, it must be considered that this fact might negatively affect the duration and intensity of the analgesic action [27]. Further pharmacodynamic research is needed to determine what concentration of local anaesthetic would provide the best balance between the distribution, duration, and intensity of the block.

When a fascial plane block is performed, the administration of a volume test is a common practice used to assess whether the tip of the needle is in the target anatomical structure [28]. Due to the small size of our cat cadavers, our volume test was 0.1 mL. This small amount of injectate was chosen because the total volume employed in our study was low (range 0.62–2.0 mL), and a higher volume test would dramatically reduce the remaining volume. The anechoic pocket generated during the volume test quickly disappeared, making it difficult to ensure the correct needle position. Even more, the anechoic pocket was not observed in some of the injections (26.09%). These findings could be due to the fast distribution of such a small volume in the DSP, helped by the pressure exerted by the probe on the thoracic wall and the reflux of injectate throughout the needle path. Despite these difficulties, the distribution of the injectate was performed in the target plane based on the CT images and the anatomical dissection of the cadavers.

Computed tomography images revealed two distinct distribution patterns: a concentrated pool near the injection point and a widespread distribution towards the epaxial muscles. A continuous plane connecting the DSP and RDL areas was clearly identified. However, the precise reason why the injectate remained in the DSP or exhibited a dorsal distribution is unclear. The freeze–thaw process, tissue hydration, and the absence of respiratory movements could have contributed to this heterogeneous distribution between the hemithoraces of the cat cadavers. It should be noted that these distribution patterns may not occur in live individuals. A discrepancy between the distribution (number of intercostal spaces) of the contrast media and the number of RCL stained was observed. This fact could be due to the dorsal distribution of the injectate observed in some hemithoraces. The RCL in the deep plane are localised in a small area in the ventral region of the SVT muscle near its serrations. The dorsal spread of the injectate could have avoided the staining of the RCL, but the RDL were dyed instead.

Thompson et al. (1991) [29] reported that due to the overlap of innervation, a minimum of three consecutive RCL should be blocked to provide analgesia to the thoracic wall. However, five or more RCL are traditionally recommended to obtain more reliable analgesia for this region [29,30,31]. In the present study, we have registered a median of 3 (0–5) RCL stained when 0.4 mL kg^−1^ of injectate was administered per hemithorax. Therefore, an increased volume or a greater number of injection points would be necessary to enhance the spread of injectate and nerve staining. Nevertheless, no studies comparing the analgesic effects of three versus five blocked nerves have been published.

Cadaveric studies of the ultrasound-guided serratus plane block in dogs reported the distribution of injectate between R1-R9 and R1-R5 in the superficial [13] and deep plane [14], respectively. However, in these studies the number of RCL stained was not reported, making impossible the comparison with our results. We observed staining of the RCL in both planes, with more consistent dyeing in the targeted deep plane. The thoracicus longus nerve, which provides innervation to the SVT, was also stained in 17.39% of hemithoraces in our study, compared to 88.5% in superficial serratus plane blocks in dogs [13]. This nerve is involved in inspiratory movements [1], and its block could impair respiratory function in certain clinical scenarios. Thus, the DSP block could be advantageous in preserving respiratory function more efficiently. Our study revealed several unexpected findings. First, some RCL were stained in the superficial plane (between SVT and LD). We observed by ultrasound that some injectate volume could have flown backwards through the needle path and reached the superficial serratus plane, where it stained some RCL. Second, the DSP approach allowed an unexpected dorsal distribution of injectate in some hemithoraces, reaching the epaxial muscles. This extended distribution resulted in the staining of the RDL. This fact could have been assisted by the dorsal recumbency of the cats during CT and anatomical dissection. More studies increasing the administered volume of injectate would be needed to find out if the RDL could be more consistently stained. If so, the DSP block would be an interesting option to induce analgesia over a more extensive area, being useful in surgeries involving the skin of the cranio-dorsal aspect of the thorax. Third, in the cadaver number 4, no nerves were found stained. This could be due to the small size of this animal (2.56 kg) and the low BCS (3/9) compared with the needle bevel size, which could have impeded the correct distribution of the injectate [32]. Fourth, in cadaver number 3, more RCL nerves were found stained in the superficial plane than in the DSP. This may be due to an incomplete perforation of the SVT, despite appropriate ultrasonographic visualisation of the needle tip, leading to a more superficial distribution of the injectate.

The DSP block has shown promising results as part of a multimodal analgesic plan in several case reports. In a cat submitted for lobectomy and pericardiectomy [19], the patient did not need rescue analgesia in the postoperative period. In dogs, two case report series performing the superficial serratus plane block [17] or DSP block [18] for PDA surgical correction have been published. Both techniques provided analgesia to the thoracic wall. In another case report [16], a superficial serratus plane block was performed in 4 dog thoracotomies. The animals did not show any nociceptive episodes in the intraoperative period. Furthermore, the combination of the DSP block and the transversus abdominis plane block has also shown analgesic properties in mastectomies in dogs [15]. Despite all these case reports, clinical studies are still lacking. Other locoregional techniques have been developed to provide analgesia to the thoracic wall, such as thoracic epidural [33], erector spinae plane block [34], thoracic paravertebral block [35], and intercostal nerve block [36]. Despite this information, it remains to be determined which of these techniques provides better analgesia in each clinical situation.

Our study has several limitations. This study was performed in cadavers, and the process of freezing and thawing could have altered the muscular and ligament forces and tensions, the hydration status, and the ultrasound image features. The dorsal recumbency position, in which the cadavers were placed during the CT scan, could also exert some influence on the distribution of the injectate. In addition, the muscular movement during spontaneous or mechanical ventilation may alter the distribution pattern of the injectate. Even more, the physicochemical characteristics of methylene blue and iopromide compared to commonly used local anaesthetics could have also altered the spread of the mixture [37]. Due to the small size of the animals and the structures involved in this technique, the unknown learning curve [38] of this block may require a larger number of animals to enhance the effectiveness of the DSP block. Finally, the researcher who performed the ultrasound technique and the anatomical dissection was the same, and it could have biassed the results.

## 5. Conclusions

Our study shows that the DSP block in cat cadavers is a feasible technique. It regularly stained the RCL of the intercostales nerves of T4 to T6 and, less frequently, T2, T3, and T7. Our results show that, integrated into a multimodal analgesic protocol, this locoregional block could be useful in craniolateral thoracotomies in cats. The DSP block also frequently dyed the RDL of the rami dorsales from T2 to T7, being capable of, theoretically, extending the desensitised area to the skin of the dorsum. Further studies are necessary to assess the analgesic properties of this block in a clinical scenario.

## Figures and Tables

**Figure 1 animals-14-02978-f001:**
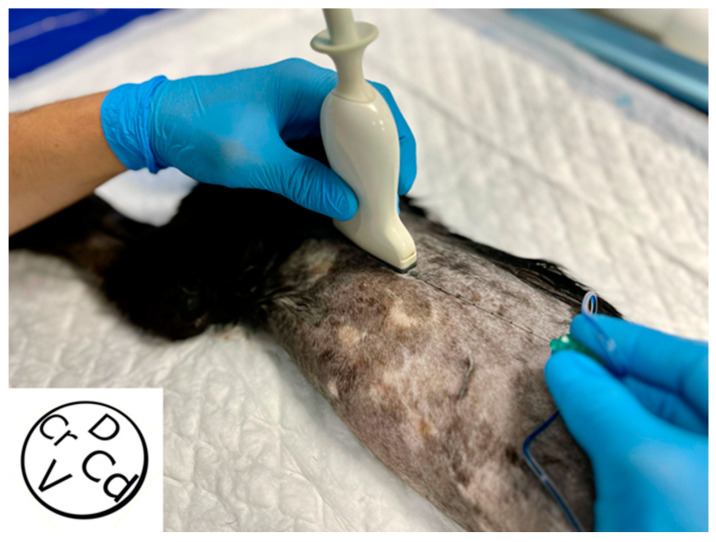
Ultrasound probe position to perform the deep serratus plane block. The probe is positioned at the level of the acromion, transversal to the ribs. The needle is advanced “in plane” in a caudo-cranial direction. D, dorsal; V, ventral; Cr, cranial; Cd, caudal.

**Figure 2 animals-14-02978-f002:**
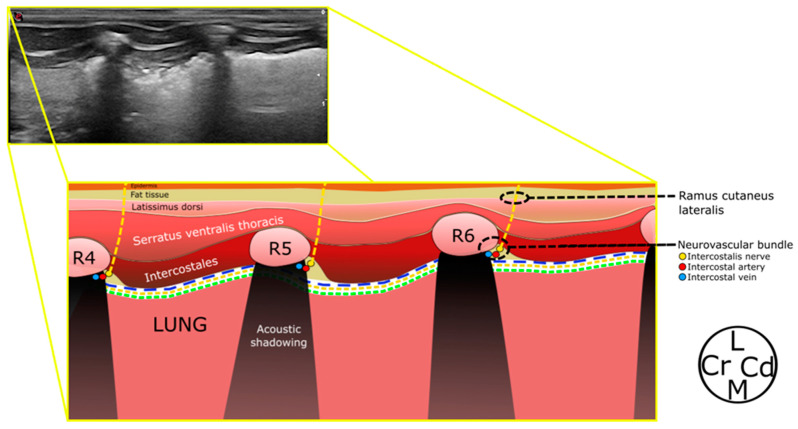
Schematic representation of the ultrasound image of the lateral aspect of the thoracic wall. The rami cutanei laterales emerge from the intercostales nerves and travel laterally towards the epidermis. The ribs produce an acoustic shadowing and the endothoracic fascia (blue dotted line), parietalis (orange dotted line), and visceralis (green dotted line) pleuras are seen as a hyperechoic line. R4–R6, ribs 4–6; L, lateral; Cr, Cranial; Cd, Caudal; M, medial.

**Figure 3 animals-14-02978-f003:**
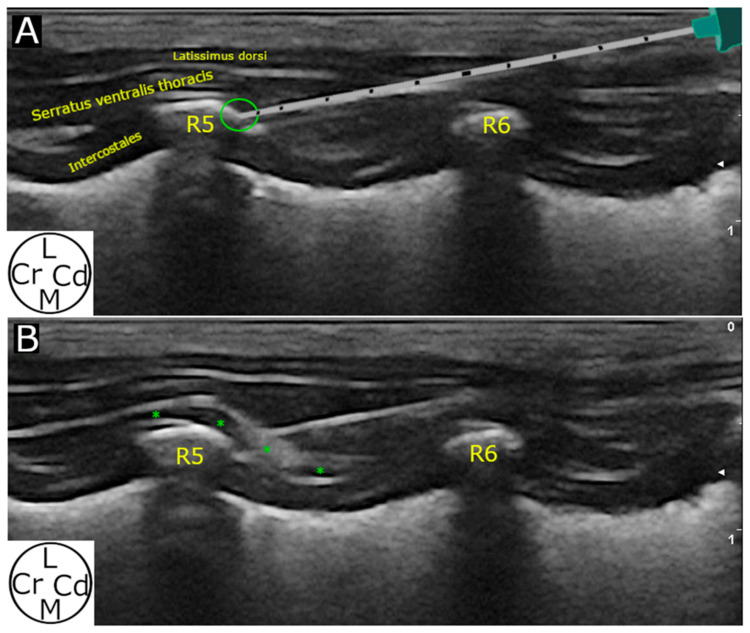
Ultrasound imaging and execution of the deep serratus plane block. (**A**) The needle is advanced “in plane” until contact with the R5 at its caudo-lateral border. (**B**) An anechoic pocket (green asterisks) is formed after injecting the mixture of methylene blue and iopromide between the serratus ventralis thoracis muscle and the ribs/intercostales externi muscles. R5, fifth rib; R6, sixth rib; L, lateral; Cr, cranial; Cd, caudal; M, medial.

**Figure 4 animals-14-02978-f004:**
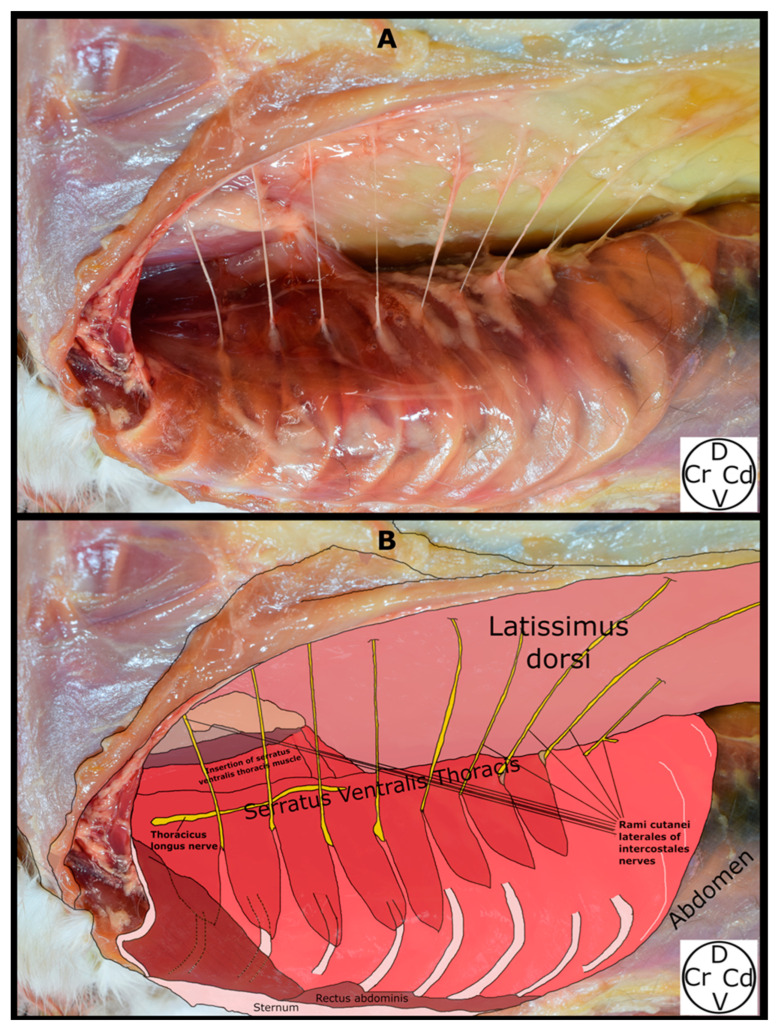
Superficial plane of the serratus ventralis thoracis (SVT) muscle in a cat cadaver dissection. The scapula and superficial muscle layers lateral to the SVT have been lateralized. The rami cutanei laterales of the intercostales nerves are observed between the serrations of the SVT. (**A**) Anatomical visualisation. (**B**) Schematic drawing of the anatomical structures. D, dorsal; Cr, cranial; Cd, caudal; V, ventral.

**Figure 5 animals-14-02978-f005:**
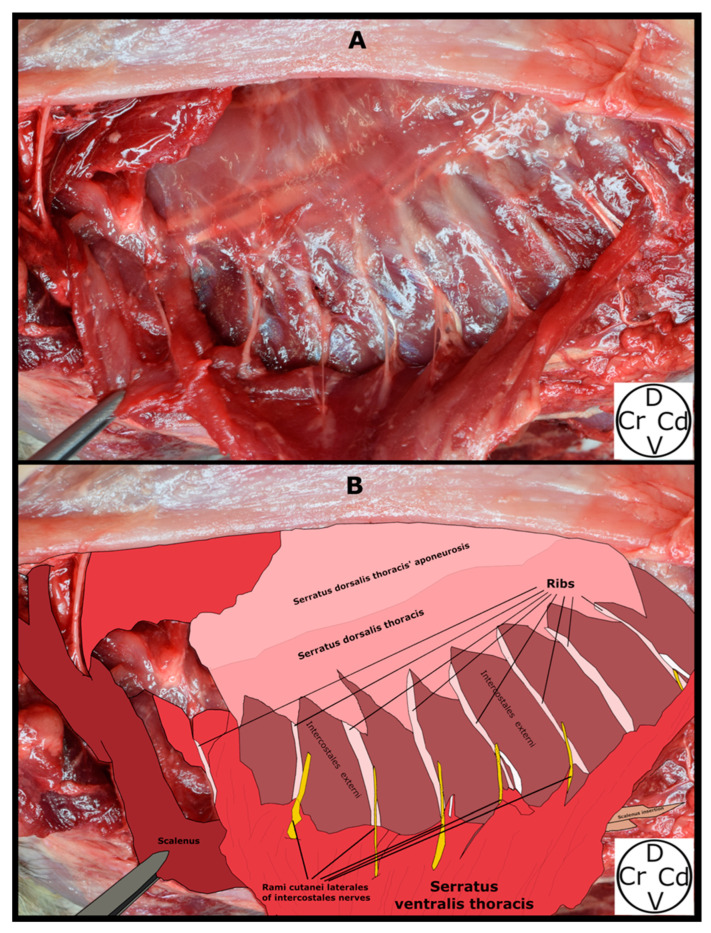
Deep plane of the serratus ventralis thoracis muscle (detached from its insertion) in a cat cadaver dissection. The rami cutanei laterales are seen emerging caudal to their respective ribs. (**A**) Anatomical visualisation. (**B**) Schematic drawing of the anatomical structures. D, dorsal; Cr, cranial; Cd, caudal; V, ventral.

**Figure 6 animals-14-02978-f006:**
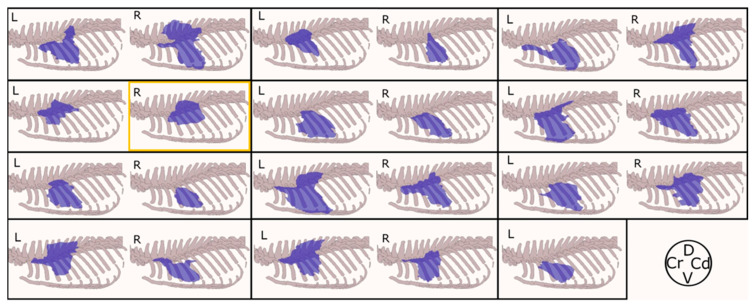
Contrast media distribution pattern of the 23 hemithoraces subjected to CT scan after performing the ultrasound-guided deep serratus plane block with 0.4 mL kg^−1^ of a mixture of iopromide and methylene blue. Twenty-three hemithoraces are represented in purple by their respective distribution patterns. In one hemithorax (yellow square), contrast was found in the subcutaneous space. L, left hemithorax; R, right hemithorax; D, dorsal; Cr, cranial; Cd, caudal; V, ventral.

**Figure 7 animals-14-02978-f007:**
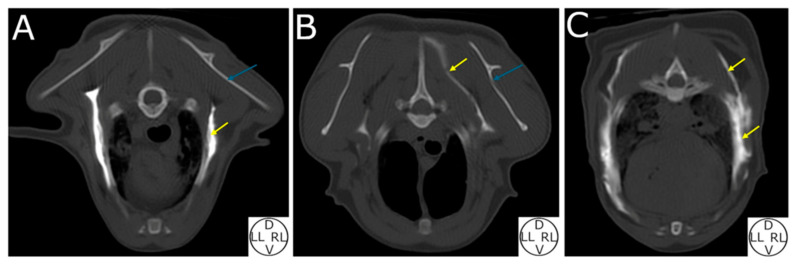
Computed tomography transverse images of three cat cadavers after performing an ultrasound-guided deep serratus plane block. (**A**) Image at the level of the fifth thoracic vertebra. Contrast media pooled in the target area between the serratus ventralis thoracis and intercostales externi muscles. (**B**) Image at the level of the fourth thoracic vertebra. Contrast media in the right hemithorax reach the epaxial muscles. (**C**) Image at the level of the sixth thoracic vertebra showing a combination of A and B patterns. Yellow arrow, contrast media; blue arrow, scapula; D, dorsal; LL, left lateral; RL, right lateral; V, ventral.

**Figure 8 animals-14-02978-f008:**
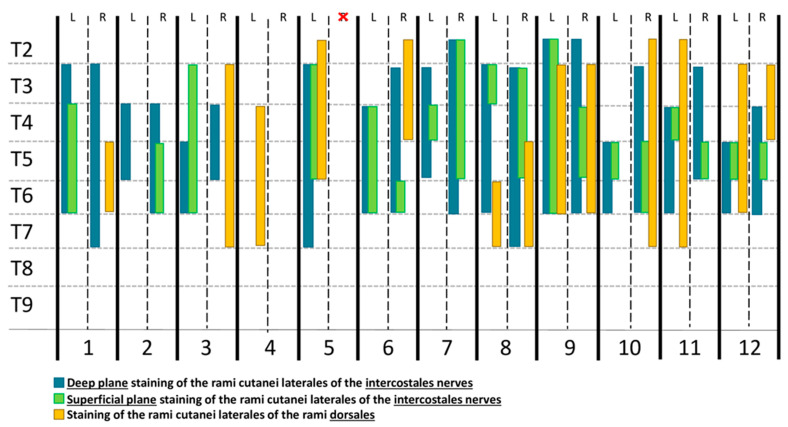
Diagram of the number of nerves stained after performing the deep serratus plane block with a mixture of 0.4 mL kg^−1^ of methylene blue and iopromide 50:50 in 12 cat cadavers. One hemithorax was excluded (red cross). L, left hemithorax; R, right hemithorax; T2, T3, T4, T5, T6, T7, and T9, spinal nerves; 1–12, cats 1–12.

**Figure 9 animals-14-02978-f009:**
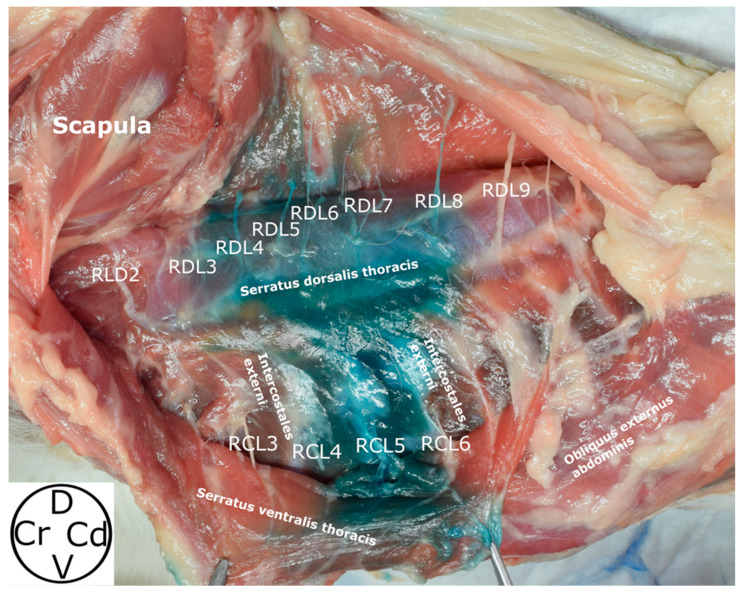
Distribution of dye in the deep serratus ventralis plane after the ultrasound-guided injection of a mixture of methylene blue and iopromide 50:50. RDL, rami laterales of the rami dorsales of the spinal nerves; RCL, rami cutanei laterales of the intercostales nerves; D, dorsal; Cr, cranial; Cd, caudal; V, ventral.

**Table 1 animals-14-02978-t001:** Number and percentage of RCL stained in the cat cadavers either in the deep or the superficial plane of the serratus ventralis thoracis following DSP injection of a 0.4 mL kg^−1^ mixture of iopromide and methylene blue. RCL; rami cutanei laterales.

	Location of the Staining	
Nerve	Deep Plane	Superficial Plane
RCL2	17.39% (4/23)	8.70% (2/23)
RCL3	52.17% (12/23)	26.09% (6/23)
RCL4	78.26% (18/23)	43.48% (10/23)
RCL5	91.30% (21/23)	65.22% (15/23)
RCL6	78.26% (18/23)	26.09% (6/23)
RCL7	8.70% (2/23)	0% (0/23)
RCL8	0% (0/23)	0% (0/23)

## Data Availability

Data supporting the reported results can be sent to anyone interested by contacting the corresponding author.

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
