# Peer review of "Ultrasound-Guided Deep Serratus Plane Block in Cat Cadavers (Felis catus): A Description of Dye and Contrast Media Distribution"

_animals, 2024, doi:10.3390/ani14202978_

Round 1
Reviewer 1 Report
Comments and Suggestions for Authors
The manuscript is clear, the information is presented in an orderly manner and provides valuable information in the field of interfascial blocks in felines.
The cites references are adequate in number and dates. No self-citations were detected.
The experimental design of the manuscript is appropriate since it provides information from three perspectives: the performance of the regional block, the computed tomography, and the analysis of the anatomy using stains.
The results are reproducible.
It presents very good quality images where the anatomical references are clearly observed. The graphics are adequate and clear. The statistical contribution is simple, but adequate.
The conclusions are appropriate and provide interesting theories linked to the dorsal migration of the anesthetic, which could be of great clinical value.
In summary, the manuscript provides valuable information in the field of thoracic interfascial regional blocks in felines.
Author Response
Thank you for your time reviewing this submission, we appreciate your positive evaluation and we are glad you found it satisfactory.
Reviewer 2 Report
Comments and Suggestions for Authors
Thank you for submitting the manuscript for review. The paper has a good experimental design and is well executed. The results obtained are interesting for the study and understanding of the serratus block's behavior in the thoracic region of the cat.
The images are of high quality. I believe that after addressing these minor comments, the article will be suitable for publication.
I suggest subdividing Figure 1 into parts (a) and (b) and adjusting the figure legend accordingly.
For Figure 4, I suggest subdividing it into four parts, so that each figure has a letter, and adjusting the figure legend according to this change.
Line 181 - Please indicate the reference used for considering 1 cm in the dye assessment.
Given the presented results, it may be important to specify in the Materials and Methods, between lines 178-185, whether the presence of dye was evaluated in the superficial or deep fasciae.
I suggest deleting lines 296-297, as the superficial approach is not discussed, and it does not seem relevant to include this sentence in the discussion.
I believe the paragraph between lines 296-304 can be rewritten, as both approaches to the serratus are mentioned, while this study only evaluates the deep approach.
Line 314-315: “Therefore, the DSP block could potentially cover more RCL in cats than in dogs.” Why do you state this? What is it based on? Please cite or explain
Lines 318-321 - The study is not an in vivo study, so it does not seem necessary to address this matter in a dispersion evaluation study.
Lines 321-323 - Remove, as it is already mentioned in lines 353-355.
Lines 377-379 - Please compare the success rate of your study with the success rate of the serratus block in dogs.
Include in the study limitations the fact that the dorsal recumbency position, after block performance, may have influenced the distribution, as previously mentioned.
Line 408 - I suggest removing “being able to provide analgesia to the cranio-lateral thoracic wall.” This can only be stated in a clinical study.

It only requires slight improvements, especially in the discussion.
Author Response
We sincerely appreciate your time and effort in reviewing my submission. Your comments and suggestions were very helpful.
Remark: I suggest subdividing Figure 1 into parts (a) and (b) and adjusting the figure legend accordingly.
Answer: Thank you for your suggestion, as the bottom left image did not provide any new information it has been removed, and the compass has been modified (following other reviewer’s remarks)
Remark: For Figure 4, I suggest subdividing it into four parts, so that each figure has a letter, and adjusting the figure legend according to this change.
Answer: Thank you for your suggestion. To see clearly the 4 images, we have decided to separate them into two figures. We hope you consider adequate this modification. The rest of the figures have been renumerated
Remark: Line 181 - Please indicate the reference used for considering 1 cm in the dye assessment.
Answer: The reference is now included.
Remark: Given the presented results, it may be important to specify in the Materials and Methods, between lines 178-185, whether the presence of dye was evaluated in the superficial or deep fasciae.
Answer: Information added
Remark: I suggest deleting lines 296-297, as the superficial approach is not discussed, and it does not seem relevant to include this sentence in the discussion.
Answer: Even though the superficial plane of the SVT was not our target plane, we consider important the inclusion of this information because we also assess the presence of dye in the superficial plane, as is now stated in material and methods
Remark: I believe the paragraph between lines 296-304 can be rewritten, as both approaches to the serratus are mentioned, while this study only evaluates the deep approach.
Answer: The assessment of the superficial plane is now included in the M&M section. Thank you for your appreciation.
Remark: Line 314-315: “Therefore, the DSP block could potentially cover more RCL in cats than in dogs.” Why do you state this? What is it based on? Please cite or explain
Answer: The serratus ventralis extends one rib more caudal, this hypothetically could have led to a more extended distribution of the injectate, reaching one more nerve
Remark: Lines 318-321 - The study is not an in vivo study, so it does not seem necessary to address this matter in a dispersion evaluation study.
Answer: We agree. This is not an in vivo study. However, this study serves as a starting point to understand the distribution of a injectate in the DSP which could help to improve and understand the technique and its clinical outcomes in a clinical scenario. That is why we consider important to consider this fact.
Remark: Lines 321-323 - Remove, as it is already mentioned in lines 353-355.
Answer: removed
Remark: Lines 377-379 - Please compare the success rate of your study with the success rate of the serratus block in dogs.
Answer: Unfortunately the nerves stained in the dog’s article were not reported, so direct comparison between species and studies cannot be accurately performed. This is stated in the discussion.
Remark: Include in the study limitations the fact that the dorsal recumbency position, after block performance, may have influenced the distribution, as previously mentioned.
Answer: Added
Line 408 - I suggest removing “being able to provide analgesia to the cranio-lateral thoracic wall.” This can only be stated in a clinical study.
Answer: This statement has been rephrased
Reviewer 3 Report
Comments and Suggestions for Authors
Dear Authors,
Thank you for submitting this observational cadaveric study to Animals.
The ultrasound-guided serratus plane block has not been previously investigated in cats; therefore, this study provides some information that may help improve the technique in this species.
Please see my comments on the manuscript below.
General comments:
Several abbreviations have been used, making the paper difficult to read, as remembering all of them is challenging. I strongly suggest reporting in full those names that are rarely used (i.e., II, IE, PP, PS, PDA, LD...). I would maintain only DSP, SVT, RCL, and RDL.
The abbreviation R has been used to indicate rib and right. I suggest using it only for “right.” When used to abbreviate rib, it was often accompanied by the word “rib” anyway (for example, in lines 119 and 199).
Even though 24 injections were performed, data from one injection was not included in the analysis. The authors did not explain the exact location of the “wrong” injection and the reason for exclusion. Conversely, later in the result section, the authors reported that a second injection was (wrongly) done in the subcutaneous tissue; these data were not excluded from the analysis. I would suggest better explaining how the authors confirmed the two “wrong” injections and the rationale behind the inclusion or exclusion of the data.
The authors decided to inject a volume of 0.4 ml/kg, stating that this would correspond to a dose of 2 mg/kg of bupivacaine or ropivacaine 0.5%. However, this volume/concentration of anesthetic calculation is correct only if the block is performed unilaterally. In this study, the authors performed the DSP bilaterally, using a total volume of 0.8 ml/kg, which would lead to a dose of 4 mg/kg of bupivacaine 0.5%. I suggest the authors further discuss the need to employ a more diluted solution to perform the DSP bilaterally or in combination with other blocks (i.e. TAP block for radical mastectomy).
The authors also stated that dilution of local anesthetic will “sacrifice intensity and duration” of the blockade. Even though I do not disagree, I would highlight that fascial plane blocks rely on large volumes of diluted anesthetic. Moreover, the quality of postoperative pain control provided by TAP block using levobupivacaine administered as either a low volume/high concentration or a high volume/low concentration solution did not differ (Sola et al. 2019; DOI: 10.1213/ANE.0000000000003736 ). To date, we lack pharmacokinetic and pharmacodynamic studies on local anesthetics administered via fascial plane blocks in veterinary medicine. However, in the case reports on the DSP block mentioned by the authors, diluted solutions of local anesthetics were employed (ropivacaine 0.16% in a cat, bupivacaine 0.125% and levobupivacaine 0.15% - 0.125%, in dogs).
Finally, the conclusions of both the abstract and main document are misleading as the RCL nerves were not consistently stained, nor could the analgesic effect of this block be assessed. Only a small portion of RCL nerves were stained, and always in less than 100% of injections. Therefore, in a clinical scenario, this LRA technique with local anesthetic may desensitize only the superficial muscle and skin corresponding to 2-3 intercostal spaces without desensitizing the intercostal muscles.
Simple Summary:
Line 16: The “lateral thoracic wall” may be confusing because the RCLs innervate only the superficial muscular layers and skin of the ventro-lateral hemithorax.
Line 17: “Thoracic” should be spelled in full and then abbreviated as “T.”
Lines 15-16: The conclusion sentence is misleading because the injectate spread in a limited area of the hemithorax. The nerves RCL2 and RCL7 were occasionally stained. Moreover, this is a cadaveric study, and the analgesic effect of this locoregional technique with a local anesthetic was not investigated. I suggest limiting your conclusion to the anatomical findings.
Abstract:
Line 28: as mentioned earlier, “lateral” is a generic definition (the intercostal nerves are unaffected by a DSP blockade).
Line 38: For consistency, remove any space between RCL and the number (i.e., RCL2).
Please add “within the DSP,” as only the % of RCL stained in the target plan is reported here.
Lines 39-41: If possible, I would add that occasionally, the dorsal cutaneous branches and the thoracicus longus nerve were stained.
Line 40: Please add something like “with a volume of anesthetic similar to the one used in this study” after “the DSP block.”
“Cranio-lateral thoracic wall” – please see comment on line 16.
Introduction:
Lines 49-50: delete T2 and T12, and add the abbreviation (T) after thoracic.
Lines 50-56: Drs. Portela and Romano (Ref.3) described the anatomy of the thoracic wall in dogs. Although the anatomy could be similar in cats, this needs to be stated. Otherwise, please provide a reference for cats.
Lines 54, 58, and 64: Please remove the abbreviations II, IE, and PDA (see general comment regarding abbreviations).
Line 56: Please name the thoracic lateral (superficial) muscles you are referring to.
Line 60: As mentioned before “cranio-lateral thoracic wall may be imprecise.”
Read (Ref.5) refers to the “ventrolateral hemithorax.”
Line 70: Please state “as part of a multimodal analgesic protocol.”
Line 74: please change “using” to “following” or “injected by.”
Line 75 “feasible in cats and a volume of 0.4 ml kg-1 of injectate would stain (...)”
Materials and Methods:
Lines 90-91: please remove the abbreviations R1 and R13.
Lines 93-94: please remove the abbreviation PS and PP.
Lines 106-109: Since the echogenic needle employed has an extension line, I assumed more than 0.4 ml/kg per syringe was prepared. Please rephrase.
Please also add a reference to support the mixture used.
Line 111: delete “R5”.
Was the fifth rib identified by palpation or by US? Please clarify.
Line 115: Figure 1. If the cat is in lateral recumbency, as stated in the M&M, the labels "dorsal and ventral" are incorrect.
Lines 119-120: Delete R4, R5 and R6.
Maybe consider rephrasing to explain that the sonographic landmarks to perform the DSP block were …
Line 133: Please explain why a 100 mm needle was used rather than a shorter one (as employed in the mentioned case reports).
Line 135. Please explain that “0.1 ml of the prepared solution” was used to perform an injection test.
Line 137. I assume you injected the remaining volume to reach 0.4 ml/kg.
Lines 139-140: You may consider “based on the ultrasonographic visualization of the needle tip in contact with the margin of the fifth rib.”
Lines 161-165: The deep serratus plane was abbreviated in the previous figure; you could use “DSP.” Then, change the green dots to “asterisks.”
Lines 179-181: you may shorten the two sentences as “Promptly after CT scan, cadavers were dissected following the procedure described above (2.1. phase 1. Anatomical study)”.
Lines 181-183: This part is confusing. In the Abstract, you only reported the RCL nerves. When describing the serratus block, you mentioned that it aims to desensitize the RCL nerves. Finally, the hypothesis is that the technique would affect the RCL; however, in the M&M, you assessed all nerves stained.
Although I agree that it is important to report all nerves affected by the injection (RCL, RDL, and thoracicus longus nerve), I suggest explaining what your primary outcome was (i.e., staining of target nerves RCLs within the DSP) and that you also assessed all nerve stained by the injectate within non-target planes.
Results:
Line 222: “weighed 3.33 ± 1.30 kg”
Line 225: The sentence about the needle path is a little strange because visualization of the needle pathway is the gold standard while performing an US-guided LRA technique. Moreover, one injection was excluded because dorsal to the target region and a second injection was done subcutaneously despite needle visualization in 24/24 injections. Please clarify.
Line 226: “the injectate was administered dorsal to the target region and was excluded from the study” - was it identified with the US during injection, with the CT scan or after dissection?
If the injectate was observed “dorsal to the target plane” while performing the injection, why wasn't the needle redirected to the target plane?
Could you please clarify what you mean by "dorsal to the target region"? The target injection point is the DSP at the 5th rib; do you refer to the target fascial plane?
Lines 230-232: I think it is implicit to say that the anechoic pocket was not observed in the remaining DSP injections. The sentence could be removed.
It is unclear if the 6/23 injections in which the solution refluxed around the needle correspond to all the injections in which the pocket formation did not occur.
Line 236: As mentioned earlier, the target region was not defined. Do you mean “target plane”?
Line 241: “Figures 5 and 6.”
Lines 242-246: Figure 5: you might consider adding the cat identification number in each rectangle. Doing that allows the reader to compare the spread observed at CT with the findings at dissection (figure 8).
Line 243: ultrasound-guided DSP block with 0.4 ml kg-1 of a mixture of...
Lines 252-258: DSP was already abbreviated.
The blue arrows are difficult to visualize in the figures; I suggest using white arrows.
For consistency, use the labels L and R (not LL and RL).
Line 260: You might start the sentence by explaining where the injectate was localized (i.e. in the DSP following 22/23 injection… in 1/23 was subcutaneous… dye solution was also found in the plane superficial to the SVT muscle…). Then you could report the nerves successfully stained (RCL in DSP first).
Line 260: A median of 3 (0-5) RCL nerves (…) were stained in the target DSP, whereas a median of 1 (0-5) RCL nerves were stained in the plane superficial to the muscle SVT.
Line 262: (Table 1 and Figure 8).
Please note that the figures may need to be renumerated.
Please add a sentence to report no intrathoracic/pleural injection (if so).
Lines 267-268: You may want to add “following DSP injection of a mixture of iopromide and methylene blue in 12 cat cadavers (23 hemithorax)”.
Line 288: Figure 8: In cat 3, left hemithorax, more RCL nerves were stained in the superficial plane than in the DSP. Please comment on that in the discussion.
In cat 4, one injection was SC, and the other did not stain any RCL; please comment on the discussion with any possible explanation.
Discussion:
Line 293: I would not say that the RCL nerves from 4 to 6 were consistently stained. Even if we combine the nerves stained either in the deep or superficial plane, RCL4 was stained in the 19/23 injection (82%), RCL5 in 21/23 (91%), and RCL6 in 17/23 (74%)—if we also consider the failed injection, the incidence of nerve staining is even lower.
Lines 294-295: Please rephrase because the sentence is unclear. Only a small portion of the superficial cranio-lateral thoracic wall will be desensitized by an anesthetic solution.
Lines 296-297: in dogs.
Lines 297-299: This is a cadaveric study; you are not assessing the analgesic effect of the locoregional technique. Please use verbs like "to affect" or "to stain."
Line 308: “middle section” is unclear. Do you mean in the middle length of the muscle?
Line 310: “a species anatomically close to cats”; this should be stated in the introduction when you report the description of the canine thoracic wall.
Line 316: RCL, no RLC
Lines 316-322: These sentences sound a little awkward. Please see my general comment on the volume and concentration of the anesthetic.
Please note that if the block is performed bilaterally or in combination with another locoregional technique, the anesthetic must be diluted to respect the maximum recommended dose.
Please consider rephrasing (i.e., a larger anesthetic volume could potentially result in wider cranial-to-caudal spread, but further dilution of anesthetic may affect the blockade intensity and duration. Further studies are needed to…).
Lines 334-335: I would delete this sentence because you did not clarify what "adequate" means. Moreover, the cranio-caudal spread in proximity to the injection point seams limited (the solution spread in different planes—more superficial and dorsal—as you explain later).
Lines 357-363: I would be careful comparing two species and different blocks. Please limit the discussion to your findings (e.g., even though the thoracic longus nerve was stained in only 17% of cases, impairment of the inspiratory muscles is a potential complication of this LRA technique).
Line 379: I wonder why reference 28 is relevant. A cadaveric study on TAP block in cats also stated that the plane block was difficult to perform in cats < 2 kg due to the small size of the plane and the needle dimension (Garbin et al. 2022; DOI: 10.3390/ani12192674 ).
Lines 380-392:
The paragraph sounds like a repetition of the information presented in the introduction without apportioning any new information. I would simply state the need for clinical trials to assess the DSP's analgesic efficacy in cats. If you decide to maintain the paragraph, please comment that in the cases reported, the DSP was always used as part of a multimodal analgesic plan (its analgesic efficacy has yet to be determined).
Line 400: add the reference (De Miguel Garcia et al. 2020; https://doi.org/10.1016/j.vaa.2020.01.003)
Conclusion:
Lines 406-408: Please be careful with the conclusion. The volume and technique studied did not consistently stain the RCL4 to 6. The nerves RCL 2, 3 and 7 were occasionally stained (no less frequently). Moreover, you did not assess the analgesic properties of the technique (remove “being able to provide analgesia to the cranio-lateral thoracic wall”).
This study may help design further studies and implement the technique in cats.
Comments on the Quality of English Language
Based on my English proficiency, the article's English level is adequate. However, editing by a native English speaker might help formulate some sentences more effectively and concisely.
Author Response
We appreciate your thorough review and valuable feedback. Your insights have significantly contributed to improving the quality of this work.
General comments:
Remark: Several abbreviations have been used, making the paper difficult to read, as remembering all of them is challenging. I strongly suggest reporting in full those names that are rarely used (i.e., II, IE, PP, PS, PDA, LD...). I would maintain only DSP, SVT, RCL, and RDL.
Answer: Thank you for your remark, we have deleted all abbreviations except IE and LD as they are mentioned several times in the text
Remark: The abbreviation R has been used to indicate rib and right. I suggest using it only for “right.” When used to abbreviate rib, it was often accompanied by the word “rib” anyway (for example, in lines 119 and 199).
Answer: We have deleted all “R” abbreviations that refers to “rib”
Remark: Even though 24 injections were performed, data from one injection was not included in the analysis. The authors did not explain the exact location of the “wrong” injection and the reason for exclusion. Conversely, later in the result section, the authors reported that a second injection was (wrongly) done in the subcutaneous tissue; these data were not excluded from the analysis. I would suggest better explaining how the authors confirmed the two “wrong” injections and the rationale behind the inclusion or exclusion of the data.
Answer: Thank you for your appreciation. This subject has been clarified in the text.
Remark: The authors decided to inject a volume of 0.4 ml/kg, stating that this would correspond to a dose of 2 mg/kg of bupivacaine or ropivacaine 0.5%. However, this volume/concentration of anesthetic calculation is correct only if the block is performed unilaterally. In this study, the authors performed the DSP bilaterally, using a total volume of 0.8 ml/kg, which would lead to a dose of 4 mg/kg of bupivacaine 0.5%. I suggest the authors further discuss the need to employ a more diluted solution to perform the DSP bilaterally or in combination with other blocks (i.e. TAP block for radical mastectomy).
Answer: We appreciate your comment. The reason why this is not further discussed is that this block mainly used unilaterally. We completely agree with your comment, in bilateral surgeries, a dilution of bupivacaine should be done.
Remark: The authors also stated that dilution of local anesthetic will “sacrifice intensity and duration” of the blockade. Even though I do not disagree, I would highlight that fascial plane blocks rely on large volumes of diluted anesthetic. Moreover, the quality of postoperative pain control provided by TAP block using levobupivacaine administered as either a low volume/high concentration or a high volume/low concentration solution did not differ (Sola et al. 2019; DOI: 10.1213/ANE.0000000000003736 ). To date, we lack pharmacokinetic and pharmacodynamic studies on local anesthetics administered via fascial plane blocks in veterinary medicine. However, in the case reports on the DSP block mentioned by the authors, diluted solutions of local anesthetics were employed (ropivacaine 0.16% in a cat, bupivacaine 0.125% and levobupivacaine 0.15% - 0.125%, in dogs).
Answer: Thank you for your comment. We agree, pharmacological and pharmacodynamic research is needed to state which concentration would provide the best analgesia.
Remark: Finally, the conclusions of both the abstract and main document are misleading as the RCL nerves were not consistently stained, nor could the analgesic effect of this block be assessed. Only a small portion of RCL nerves were stained, and always in less than 100% of injections. Therefore, in a clinical scenario, this LRA technique with local anesthetic may desensitize only the superficial muscle and skin corresponding to 2-3 intercostal spaces without desensitizing the intercostal muscles.
Answer: Conclusions have been modified
Simple Summary:
Remark: Line 16: The “lateral thoracic wall” may be confusing because the RCLs innervate only the superficial muscular layers and skin of the ventro-lateral hemithorax.
Answer: Thank you for your comment. When we write “lateral thoracic wall” we are excluding the dorsum, which is inervated by the rami dorsales laterals and also the ventral thoracic wall (sternum and preriferal area) innervated by the rami cutanei ventrales. We chose “lateral thoracic wall” state the limits of the RCL innervated region.
Remark: Line 17: “Thoracic” should be spelled in full and then abbreviated as “T.”
Answer: Thank you for your comment. Abbreviation “T” is now defined as “thoracic spinal nerve”
Remark: Lines 15-16: The conclusion sentence is misleading because the injectate spread in a limited area of the hemithorax. The nerves RCL2 and RCL7 were occasionally stained. Moreover, this is a cadaveric study, and the analgesic effect of this locoregional technique with a local anesthetic was not investigated. I suggest limiting your conclusion to the anatomical findings.
Answer: Thank you, we have modified the conclusion.
Abstract:
Remark: Line 28: as mentioned earlier, “lateral” is a generic definition (the intercostal nerves are unaffected by a DSP blockade).
Answer: Answered above
Remark: Line 38: For consistency, remove any space between RCL and the number (i.e., RCL2).
Please add “within the DSP,” as only the % of RCL stained in the target plan is reported here.
Answer: Changed
Remark: Lines 39-41: If possible, I would add that occasionally, the dorsal cutaneous branches and the thoracicus longus nerve were stained.
Answer: Added
Remark: Line 40: Please add something like “with a volume of anesthetic similar to the one used in this study” after “the DSP block.”
Answer: We have added a similar sentence.
Remark: “Cranio-lateral thoracic wall” – please see comment on line 16.
Answer: Answered above
Introduction:
Remark: Lines 49-50: delete T2 and T12, and add the abbreviation (T) after thoracic.
Answer: Done
Remark: Lines 50-56: Drs. Portela and Romano (Ref.3) described the anatomy of the thoracic wall in dogs. Although the anatomy could be similar in cats, this needs to be stated. Otherwise, please provide a reference for cats.
Answer: Thank you for your appreciation. The reference requested has been added
Remark: Lines 54, 58, and 64: Please remove the abbreviations II, IE, and PDA (see general comment regarding abbreviations).
Answer: As suggested, these abbreviations have been removed
Remark: Line 56: Please name the thoracic lateral (superficial) muscles you are referring to.
Answer: Thank you for this appreciation. The sensitive innervation of the lateral thoracic musculature remains unclear. The authors understand that these branches provide innervation to the serratus ventralis, latissimus dorsi and skin. To the best of our knowledge, anatomical reports defining whether or not the sensitive innervation of the intercostal muscles is provided by the RCL are lacking. In our clinical experience, this anesthetic block technique provides a high-quality analgesia during thoracic surgeries. However, we cannot firmly state if it is due to an analgesia of the most superficial layers or if it also includes the intercostales muscles. We have rephrased the sentence.
Remark: Line 60: As mentioned before “cranio-lateral thoracic wall may be imprecise.”
Read (Ref.5) refers to the “ventrolateral hemithorax.”
Answer: Answered above
Remark: Line 70: Please state “as part of a multimodal analgesic protocol.”
Answer: Stated
Remark: Line 74: please change “using” to “following” or “injected by.”
Answer: Changed
Remark: Line 75 “feasible in cats and a volume of 0.4 ml kg-1 of injectate would stain (...)”
Answer: Suggestion added
Materials and Methods:
Remark: Lines 90-91: please remove the abbreviations R1 and R13.
Answer: Abbreviations deleted
Remark: Lines 93-94: please remove the abbreviation PS and PP.
Answer: Abbreviations deleted
Remark: Lines 106-109: Since the echogenic needle employed has an extension line, I assumed more than 0.4 ml/kg per syringe was prepared. Please rephrase.
Please also add a reference to support the mixture used.
Answer: Thank you for your comment. Yes, we prefilled needle and extension with the injectate.
The mixture of methylene blue and iopromide is the same one our research group has been used in previous publications. References are now included.
Remark: Line 111: delete “R5”. Was the fifth rib identified by palpation or by US? Please clarify.
Answer: The abbreviation was deleted, and a clarification has been added
Line 115: Figure 1. If the cat is in lateral recumbency, as stated in the M&M, the labels "dorsal and ventral" are incorrect.
Thank you for your remark. The compass indicating “dorsal and ventral” should be interpreted in 2D parallel to the table, not transversal. We have modified the figure’s compass to help clarify this subject. In addition, following another reviewer’s remark, we have deleted the small bottom left figure, as it did not add any further information.
Remark: Lines 119-120: Delete R4, R5 and R6.
Maybe consider rephrasing to explain that the sonographic landmarks to perform the DSP block were …
Answer: Abbreviations have been deleted and we have rephrased the first sentence
Remark: Line 133: Please explain why a 100 mm needle was used rather than a shorter one (as employed in the mentioned case reports).
Answer: Thank you for your comment. These long needles are employed because this study is included in a series of studies where several blocks are investigated in different species, and we use the same research material in all of them.
Remark: Line 135. Please explain that “0.1 ml of the prepared solution” was used to perform an injection test.
Answer: Clarification added
Remark: Line 137. I assume you injected the remaining volume to reach 0.4 ml/kg.
Answer: This sentence has been changed.
Remark: Lines 139-140: You may consider “based on the ultrasonographic visualization of the needle tip in contact with the margin of the fifth rib.”
Answer: This sentence has been changed
Remark: Lines 161-165: The deep serratus plane was abbreviated in the previous figure; you could use “DSP.” Then, change the green dots to “asterisks.”
Answer: The remark has been added.
Remark: Lines 179-181: you may shorten the two sentences as “Promptly after CT scan, cadavers were dissected following the procedure described above (2.1. phase 1. Anatomical study)”.
Answer: This suggestion has been added.
Remark: Lines 181-183: This part is confusing. In the Abstract, you only reported the RCL nerves. When describing the serratus block, you mentioned that it aims to desensitize the RCL nerves. Finally, the hypothesis is that the technique would affect the RCL; however, in the M&M, you assessed all nerves stained. Although I agree that it is important to report all nerves affected by the injection (RCL, RDL, and thoracicus longus nerve), I suggest explaining what your primary outcome was (i.e., staining of target nerves RCLs within the DSP) and that you also assessed all nerve stained by the injectate within non-target planes.
Answer: This sentence has been rephrased to make it clearer following your remark
Results:
Remark: Line 222: “weighed 3.33 ± 1.30 kg”
Answer: Changed
Remark: Line 225: The sentence about the needle path is a little strange because visualization of the needle pathway is the gold standard while performing an US-guided LRA technique. Moreover, one injection was excluded because dorsal to the target region and a second injection was done subcutaneously despite needle visualization in 24/24 injections. Please clarify.
Answer: Despite visualizing the needle, in one injection the solution was administered near the erector spinae region due to a mistake performing the technique. This is the reason we excluded it from the study. However, in the subcutaneous injection, the US visualization of the needle and the technique was correct, but the needle tip only pierced the subcutaneous tissue. We decided to include this last injection because the technique was correct and to raise awareness of this potential complication when performing DSP block.
Remark: Line 226: “the injectate was administered dorsal to the target region and was excluded from the study” - was it identified with the US during injection, with the CT scan or after dissection?
Answer: It was observed during CT scan and anatomical dissection
Remark: If the injectate was observed “dorsal to the target plane” while performing the injection, why wasn't the needle redirected to the target plane?
Answer: The technique mistake was noticed during CT scans
Remark: Could you please clarify what you mean by "dorsal to the target region"? The target injection point is the DSP at the 5th rib; do you refer to the target fascial plane?
Answer: Clarification has been added
Remark: Lines 230-232: I think it is implicit to say that the anechoic pocket was not observed in the remaining DSP injections. The sentence could be removed.
Answer: The sentence has been removed
Remark: It is unclear if the 6/23 injections in which the solution refluxed around the needle correspond to all the injections in which the pocket formation did not occur.
Answer: It seems independent, in 4/6 injections where the solution refluxed the anechoic pocket was formed.
Remark: Line 236: As mentioned earlier, the target region was not defined. Do you mean “target plane”?
Answer: Yes, we have modified the term
Remark: Line 241: “Figures 5 and 6.”
Answer: The number of the figures has been changed following the new order.
Remark: Lines 242-246: Figure 5: you might consider adding the cat identification number in each rectangle. Doing that allows the reader to compare the spread observed at CT with the findings at dissection (figure 8).
Answer: We understand your appreciation, however, due to its 2D nature, direct comparison with the mentioned figure could lead to confusion.
Remark: Line 243: ultrasound-guided DSP block with 0.4 ml kg-1 of a mixture of...
Answer: Your suggestion has been added.
Remark: Lines 252-258: DSP was already abbreviated.
Answer: Taking into account this abbreviation is no used in the figure, we have decided to remove the abbreviation.
Remark: The blue arrows are difficult to visualize in the figures; I suggest using white arrows.
Answer: The authors thought that white symbols could lead to confusion because CT images are in black and white.
Remark: For consistency, use the labels L and R (not LL and RL).
Answer: The letter “L” is used as “Lateral” in Figures 2 and 3, and as “left” in Figures 5 and 8. In this figure, we chose LL and RL instead of using “L” for “Left” and “R” for “Right” to avoid confusion despite inconsistency.
Line 260: You might start the sentence by explaining where the injectate was localized (i.e. in the DSP following 22/23 injection… in 1/23 was subcutaneous… dye solution was also found in the plane superficial to the SVT muscle…). Then you could report the nerves successfully stained (RCL in DSP first).
Answer: Thank you for this remark, your suggestion has been added.
Remark: Line 260: A median of 3 (0-5) RCL nerves (…) were stained in the target DSP, whereas a median of 1 (0-5) RCL nerves were stained in the plane superficial to the muscle SVT.
Answer: Your suggestion has been added
Remark: Line 262: (Table 1 and Figure 8). Please note that the figures may need to be renumerated.
Answer: Your suggestion has been added and the figures have been renumerated
Remark: Please add a sentence to report no intrathoracic/pleural injection (if so).
Answer: Sentenced added
Remark: Lines 267-268: You may want to add “following DSP injection of a mixture of iopromide and methylene blue in 12 cat cadavers (23 hemithorax)”.
Answer: The suggested clarification has been added
Remark: Line 288: Figure 8: In cat 3, left hemithorax, more RCL nerves were stained in the superficial plane than in the DSP. Please comment on that in the discussion.
Answer: Thank you for this remark, a discussion about this subject has been added.
Remark: In cat 4, one injection was SC, and the other did not stain any RCL; please comment on the discussion with any possible explanation.
Answer: Thank you for your comment, we have clarified it is cat number four
Discussion:
Remark: Line 293: I would not say that the RCL nerves from 4 to 6 were consistently stained. Even if we combine the nerves stained either in the deep or superficial plane, RCL4 was stained in the 19/23 injection (82%), RCL5 in 21/23 (91%), and RCL6 in 17/23 (74%)—if we also consider the failed injection, the incidence of nerve staining is even lower.
Answer: We have changed the term.
Remark: Lines 294-295: Please rephrase because the sentence is unclear. Only a small portion of the superficial cranio-lateral thoracic wall will be desensitized by an anesthetic solution.
Answer: This statement has been rephrased
Remark: Lines 296-297: in dogs.
Answer: Added
Lines 297-299: This is a cadaveric study; you are not assessing the analgesic effect of the locoregional technique. Please use verbs like "to affect" or "to stain."
Answer: Changed
Remark: Line 308: “middle section” is unclear. Do you mean in the middle length of the muscle?
Answer: Thank you for the remark, this description has been changed.
Remark: Line 310: “a species anatomically close to cats”; this should be stated in the introduction when you report the description of the canine thoracic wall.
Answer: The references in the introduction have been changed (as said above), therefore, stating this in the description is no longer needed.
Remark: Line 316: RCL, no RLC
Answer: Changed
Remark: Lines 316-322: These sentences sound a little awkward. Please see my general comment on the volume and concentration of the anesthetic.
Answer: Answered above
Remark: Please note that if the block is performed bilaterally or in combination with another locoregional technique, the anesthetic must be diluted to respect the maximum recommended dose.
Answer: Answered above
Remark: Please consider rephrasing (i.e., a larger anesthetic volume could potentially result in wider cranial-to-caudal spread, but further dilution of anesthetic may affect the blockade intensity and duration. Further studies are needed to…).
Answer: This sentence has been rephrased
Remark: Lines 334-335: I would delete this sentence because you did not clarify what "adequate" means. Moreover, the cranio-caudal spread in proximity to the injection point seams limited (the solution spread in different planes—more superficial and dorsal—as you explain later).
Answer: We have modified the statement
Remark: Lines 357-363: I would be careful comparing two species and different blocks. Please limit the discussion to your findings (e.g., even though the thoracic longus nerve was stained in only 17% of cases, impairment of the inspiratory muscles is a potential complication of this LRA technique).
Answer: We agree that direct comparison between species is not accurate, however, the lack of studies on cats made necessary the comparison with dogs’ block, a species anatomically close to cats.
Remark; Line 379: I wonder why reference 28 is relevant. A cadaveric study on TAP block in cats also stated that the plane block was difficult to perform in cats < 2 kg due to the small size of the plane and the needle dimension (Garbin et al. 2022; DOI: 10.3390/ani12192674 ).
We chose to use this reference as it discusses the difference in the size of the bevel and the width of the fascial planes from a histological perspective. This is even more accurate than the observations made by Garbin et al.
Remak: Lines 380-392: The paragraph sounds like a repetition of the information presented in the introduction without apportioning any new information. I would simply state the need for clinical trials to assess the DSP's analgesic efficacy in cats. If you decide to maintain the paragraph, please comment that in the cases reported, the DSP was always used as part of a multimodal analgesic plan (its analgesic efficacy has yet to be determined).
Answer: We think it is important to discuss the case reports published as they are the sole evidence of the analgesic properties of the block. We have clarified the information in the paragraph
Line 400: add the reference (De Miguel Garcia et al. 2020; https://doi.org/10.1016/j.vaa.2020.01.003)
Answer: thank you for your suggestion, we have added the reference.
Conclusion:
Remark: Lines 406-408: Please be careful with the conclusion. The volume and technique studied did not consistently stain the RCL4 to 6. The nerves RCL 2, 3 and 7 were occasionally stained (no less frequently). Moreover, you did not assess the analgesic properties of the technique (remove “being able to provide analgesia to the cranio-lateral thoracic wall”).
Answer: We have reordered and clarified this subject
Revisa muy bien el texto para que no se quede ninguna